# Framework to Measure the Mobility of Technical Talents: Evidence from China's Smart Logistics

**Jun Guan, Chunxiu Liu, Guoqiang Liang**  **and Lizhi Xing ***

School of Economics and Management, Beijing University of Technology, Beijing 100124, China
* Correspondence: koken@bjut.edu.cn; Tel.: +86-186-1016-2932

**Abstract:** Talent mobility is the key driving force to accelerate innovation and economic development. Prior studies focused much attention on the mobility of scientific talents from the angle of bibliometrics. Still, the mobility of technical talents was not thoroughly analyzed through the lens of the complex network. In consideration of technical talents being the primary and direct labor force to foster innovation and economic growth, in this paper, we provide a framework to measure the mobility of technical talents based on patents from the perspective of the complex network. The Technical Talent Mobility Network (TTMN) model is constructed to measure the changes of network topology on the levels of network, node, and edge aspects, respectively, thus deepening our understanding of the important node and mobility channels of technical talents. We then take China's smart logistics as an example to verify the framework promoted, and results show the framework can reveal the actual situation of technical talent mobility that was reported by the government gazette and related articles. The framework proposed in this paper points out a new method and perspective to measure technological talent mobility, which is essential to facilitate regional innovation and economic soar.

**Keywords:** innovation; complex network; patents; talent mobility; smart logistics

## 1. Introduction

Talent mobility can promote the spread and diffusion of knowledge, facilitate re-source integration, and bring new products and technological innovation [1–4]. As a result, how to maximize the "spillover effect" of talent mobility [5] and boost the long-term growth of enterprise and economic take-off [6] has become a hot research topic for researchers and policymakers [7].

Qualitative research and quantitative research are standard methods to explore talent mobility. The qualitative research process adopts structured [8] or semi-structured [9,10] in-depth interviews with relevant experts and analyzes the causes or results of talent mobility from psychology, economics, and other perspectives [11,12]. Compared with qualitative research, the application of quantitative research is universal and objective. Quantitative research is the main method to explore talent mobility. The existing research was used to track talent mobility through the literature databases [13–15], questionnaires [16], statistical yearbooks [17], official websites and resumes [18], etc. Then, in order to boost the innovation process in-depth and prompt economic growth, the characteristics, influencing factors [19–23], and future tendencies [24–26] of talent mobility are analyzed. For example, Jiang et al. (2022) proposed a fractional gray prediction model based on change-point detection to predict the mobility of overseas talents, and they concluded that the proportion of talents returning to China would increase steadily in the future [26]. Kongsonontornkijkul et al. (2019) constructed a theoretical framework including university factors, industry factors, research factors, etc. to explore the critical elements that influenced the participation decision of talents [27]. Robinson-Garcia et al. (2019) retrieved a dataset from 2008 to 2015 in the Web of Science database and provided a classification system to define the mobility of researchers based on their job-hopping behavior, and they found that

researchers with directionality but no rupture with their original country accounted for the largest proportion in all mobile scholars [14].

In recent years, the complex network has become a powerful tool for measuring talent mobility [28,29]. It provides an opportunity to explore the real-world system from the macroscopic level, analyzes the relationship between nodes in the network [30,31], and reveals the primary mobility law of talents [32–34]. For example, Shi et al. (2020), Wang et al. (2020), and Jin et al. (2021) measured the overall characteristics of the talent flow network by using network density, average clustering coefficient [35–37], etc. Shi et al. (2020) and Zhang et al. (2022) took advantage of degree centrality, betweenness centrality, and other indicators [35,38] to reveal the network structure and essential nodes. Based on the online database for the International Migration Statistics (IMS), Tranos et al. (2015) used the annual migration flow data of 32 OECD countries from 2000 to 2009 [39], thus exploring the importance of physical and cultural proximity to migration.

However, prior studies focused too much on scientific talents, but the mobility of technical talents needed to be thoroughly analyzed through the lens of the complex network. The consideration of technical talents is the main and direct labor force to foster technical innovation, transformation, and economic growth. In this study, technical talent mobility is investigated with the help of invention patent data. Then, by adopting complex network analysis, the framework of technical talent mobility is comprehensively constructed from the network level, node level, and edge level. It provides a reference for optimizing enterprise human resources, promoting the benign advancement of the innovation ecosystem, and accelerating the development of application-oriented fundamental research.

## 2. Methodology

### 2.1. Framework of Talent Mobility

In this section, we introduce the framework to measure talent mobility from network characteristics. By doing this, the evolution trend of the network can be comprehensively depicted and predicted. Table 1 shows the indicators and descriptions of the framework in detail.

The indicators are explained as follows:

Network density ($ND$) reflects the overall cohesion of a network and the closeness of the interconnection between nodes.

Global efficiency ($GE$) reflects the average efficiency of information transfer between node pairs, and it is equal to the harmonic mean of the distance between two nodes.

Average path length ($APL$) is often used to represent the average of the shortest edges between all nodes in the network. $APL$ describes the separation degree of nodes in the network.

Clustering coefficient ($C$) is used to gauge the degree to which nodes in a network tend to cluster together, i.e., the familiarity between nodes.

Central potential represents the degree to which a graph shows a tendency to converge to a specific node, which is used to describe the dependence of the whole network on the hub node.

Compatibility is used to measure the correlation of the degree between connected node pairs in the network.

Node betweenness centrality ($C_B$)is used to measure the transit effect of a node on the information flow between other nodes.

Edge betweenness centrality ($C_E$) is used to measure the transit effect of an edge on the information flow between other nodes.

**Table 1.** Framework for measuring talent mobility.

| Dimension | Indicator | | Formula | Explanation |
|---|---|---|---|---|
| | Network density ($ND$) | | $ND = \frac{L}{N(N-1)}$ | Where $L$ is the actual number of edges connected between vertexes, and $N$ is the number of nodes in the network. |
| | Global efficiency ($GE$) | | $GE = \frac{1}{N(N-1)} \sum_{i \neq j} \frac{1}{d_{ij}}$ | Where $N$ is the number of nodes in the network, and $d_{ij}$ is the shortest path between node $i$ to node $j$. |
| | Average path length ($APL$) | | $APL = \frac{1}{N(N-1)} \sum_{i \neq j} d_{ij}$ | Where $N$ is the number of nodes in the network, and $d_{ij}$ is the shortest path between node $i$ to node $j$. If this path does not exist in the network, it is expressed as $d_{ij} = \infty$ [40]. |
| | Clustering coefficient ($C$) | | $C(i) = \frac{A(i)}{\frac{1}{2}K(i)(K(i)-1)} = \frac{2A(i)}{K(i)(K(i)-1)}$ | Where $A(i)$ is the actual number of edges between adjacent nodes of node $i$. If node i has only one or no neighboring node (i.e., $K(i) = 1$ or $K(i) = 0$), $A(i) = 0$, and the numerator and denominator of the formula are both 0, so $C(i) = 0$. |
| Network-level | Central potential | In-degree relative central potential ($C_{RD}^{IN}$) | $C_{RD}^{IN} = \frac{\sum_{i=1}^{N}\left(C_{RDmax}^{IN} - C_{RD}^{IN}(i)\right)}{N-2}$ | Where $C_{RDmax}^{IN}$ and $C_{RDmax}^{OUT}$ are the maximum values of in-degree and out-degree relative central potential, respectively. When $C_{RD}^{OUT} > C_{RD}^{IN}$, the nodes in the network tend to connect; when $C_{RD}^{OUT} < C_{RD}^{IN}$, the nodes in the network shows the weak connection or disconnection [41]. |
| | | Out-degree relative central potential ($C_{RD}^{OUT}$) | $C_{RD}^{OUT} = \frac{\sum_{i=1}^{N}\left(C_{RDmax}^{OUT} - C_{RD}^{OUT}(i)\right)}{N-2}$ | |
| | Compatibility | In-out degree compatibility ($r_{io}$) | $r_{io} = r(\alpha, \beta) = \frac{E^{-1} \sum_i \left[\left(j_i^\alpha - \overline{j^\alpha}\right)\left(k_i^\beta - \overline{k^\beta}\right)\right]}{\sigma^\alpha \sigma^\beta}$ | Where $E$ is the total number of connected edges; $j_i^\alpha$ and $k_i^\beta$ represent the degree of source node $\alpha$ and the degree of target node $\beta$ of connected edges $i$, respectively, $\overline{j^\alpha} = E^{-1} \sum_i j_i^\alpha$, $\sigma^\alpha = \sqrt{E^{-1} \sum_i \left(j_i^\alpha - \overline{j^\alpha}\right)^2}$, and $\overline{k^\beta}$ and $\sigma^\beta$ are the same as the definition in front. When $r_{io} > 0$, the network is a collocated network, in which high-degree nodes tend to connect with nodes with a similarly higher level of degree; otherwise, it is a mismatched network, in which nodes with higher degrees tend to connect with nodes with lower degrees. |
| Node-level | Node betweenness centrality ($C_B$) [42] | Weighted betweenness centrality of node based on $RFWA$ ($C_B^{RFWA}$) [43] | $C_B^{RFWA}(i) = \sum_{i,j,k \in \{1,2,\cdots,N\}} SRPL_{jk}^{(N)}(i)$ | Where $SRPL_{jk}^{(N)}(i)$ is the number of strong relevance path length ($SRPL$) connecting any nodes pair and passing through a specific node i in the global network: $SRPL_{ij}^{(k)} = \max_{i,j,k \in \{1,2,\dots,N\}}\left\{ w_{ij}^{(k-1)}, \frac{w_{ik}^{(k-1)} w_{kj}^{(k-1)}}{w_{ik}^{(k-1)} + w_{kj}^{(k-1)}} \right\}$ Where $SRPL_{ij}^{(k)}$ is the $SRPL$ path between nodes $i$ and $j$, representing the most efficient and effective propagation path in the similarity weight network. If greater than $w_{ij}^{(k-1)}$, $SRPL_{ij}^{(k)}$ is the maximum value, or otherwise, it is only equal to $w_{ij}^{(k-1)}$. When $SRPL_{ij}^{(k)}$ happens to be equal to $w_{ij}^{(k-1)}$, the optimal path is the most direct propagation path between nodes $i$ and $j$, and thus it is unnecessary to transit through another node. |
| Edge-level | Edge betweenness centrality ($C_E$) [44] | Weighted between the centrality of edge based on $RFWA$ ($C_E^{RFWA}$) [45] | $C_E^{RFWA}(i,j) = \sum_{s,i,j,t \in \{1,2,\cdots,N\}} Str_{st}^{(N)}(i,j)$ | Where $Str_{st}^{(N)}$ is the number of links between any node pair $i$ and $j$ contained in the $SRPL$ in the whole network. |

### 2.2. Network Modeling

In this section, we illustrate the network modeling process, but before it, we run the name disambiguation process first to improve the reliability of data.

#### 2.2.1. Name Disambiguation

Technical talents in the same company may share the same names. The purpose of name disambiguation is to determine whether they are the same person after flowing. The detailed procedures are as follows:

Step 1: To cluster all subgroups of inventors with the same names. First, group patents whose inventors share the same name. In this group, patents without information about the company will be regarded as an independent subgroup and the inventors as individual applicants; companies and the inventors will group those with such information as the company applicants. All subgroups with the same names can be thus obtained.

Step 2: To set the benchmark. The patent title broadly represents the inventor's research interest. Taking the similarity of patent titles as the similarity of inventors, the average similarity of all company applicants is measured by the generalized Jaccard index. It is found that 95% of the applicants have a similarity index of no less than 0.07.

Step 3: To calculate the average similarity of applicants from different companies. The patent data in the subgroup of company applicants with the same names are paired, and then the average similarity of the applicants is calculated.

Step 4: To classify applicants from different companies in the group. Patents of company applicants are first sorted by application time. If the patents are applied by inventors with the same name but affiliated with different companies in the same time window, the inventors shall be considered diverse company applicants. If there is no time overlap, these patents need to be merged, in particular, those with the highest similarity and no less than 0.07.

Step 5: To obtain the data of technical talents after disambiguation.

Then, Step 3 and Step 4 are repeated until the similarity between all groups is less than 0.07, and name disambiguation is completed.

The process of name disambiguation is shown in Figure 1.

In the above process, we assume that the knowledge reserve and professional skills of technical talents will not change significantly, which can be used as the basis for counting the situation of talent flow between regions. This method no longer regards the organization and geographical location as the premise for identifying technical talents but judges them by the field in which they apply for patents. It lays a good foundation for the implementation of network modeling and analysis.

#### 2.2.2. TTMN Model

Complex network analysis can be widely used in sociology, physics, ecology, and so on. In recent years, this method has also been applied to the field of occupational mobility and has played an important role. Based on this, this paper constructs the TTMN model. The TTMN model is a type of weighted and directed network (WDN) model, in which: vertexes denote provinces, autonomous regions, and municipalities; edges and directions represent the flow and direction of technical talents between regions; the edge weight embodies the number of technical talents in mobility in a specific year. The total number of vertexes in the network is recorded as $N$, so the vertex set is $V = \{V_1, V_2, \cdots, V_N\}$; the edge set is $E = \{e_{11}, e_{12}, \cdots, e_{ij}, \cdots, e_{N(N-1)}, e_{NN}\}$; the weight set is $W = \{w_{11}, w_{12}, \cdots, w_{ij}, \cdots, w_{N(N-1)}, w_{NN}\}$. In the weighted network, the set of weights $W$ can be used in place of the set of edges $E$. Figure $G = (V, E, W)$ depicts the mobility of technical talents in each year. The conceptual diagram of the TTMN model is shown in Figure 2.

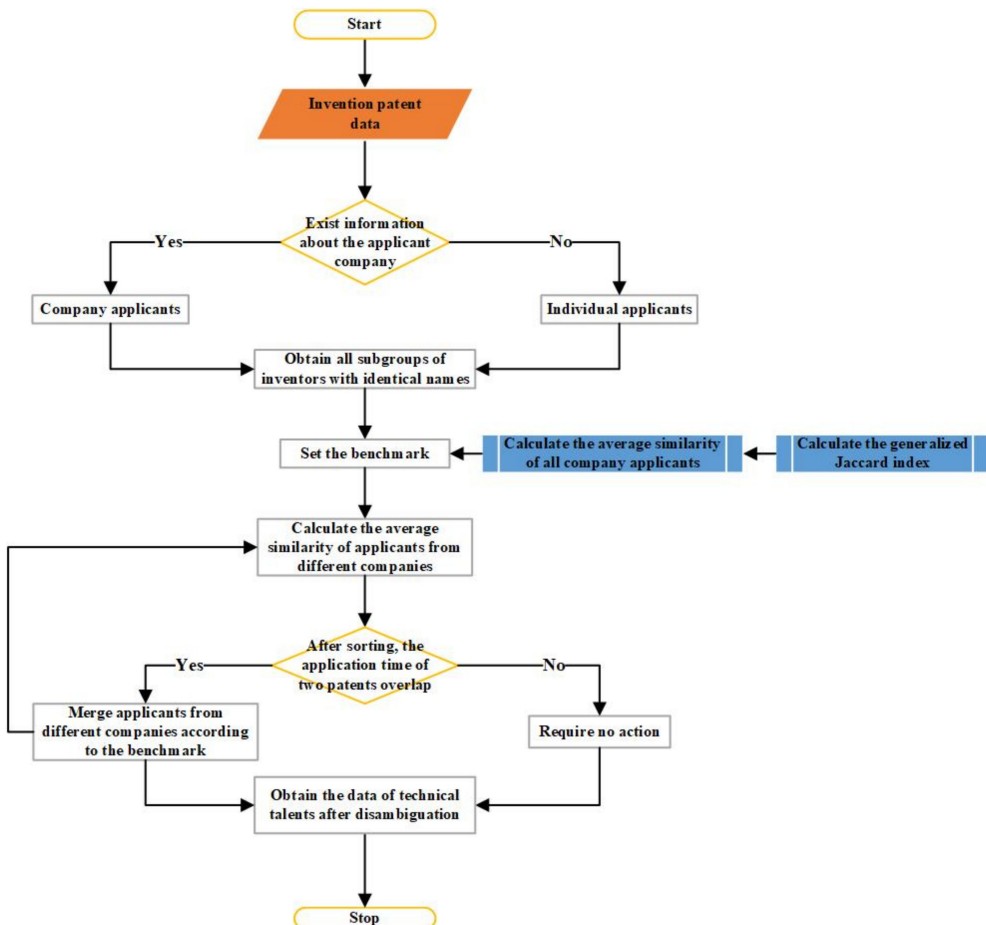

**Figure 1.** Flowchart of name disambiguation of technical talents.

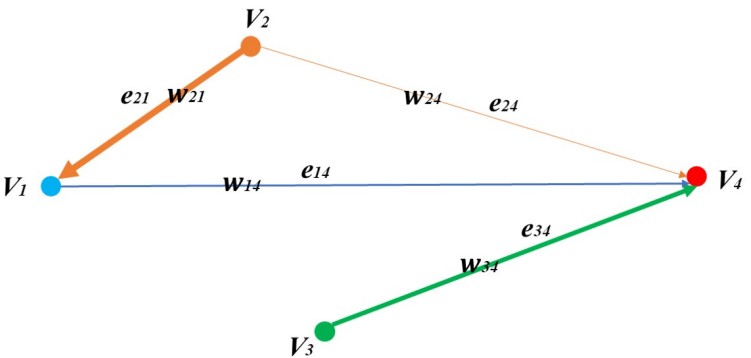

**Figure 2.** Conceptual diagram of the TTMN model.

The TTMN model is a WDN model with four nodes and four edges. The vertex set is $V = \{V_1, V_2, V_3, V_4\}$; the edge set is $E = \{e_{14}, e_{21}, e_{24}, e_{34}\}$; the weight set is $W = \{w_{14}, w_{21}, w_{24}, w_{34}\}$; weight represents the flow scale of technical talents. Obviously, $V_2$ to $V_1$ has the largest flow scale, and $V_2$ to $V_4$ has the smallest flow scale. $V_4$ only has an inflow but no outflow of talents, while $V_2$ and $V_3$ have an outflow but no inflow of talents.

## 3. Empirical Results

In 2009, IBM introduced the concept of smart logistics for the first time, attracting widespread attention. Scholars at home and abroad have carried out a series of explorations on this emerging field, mainly focusing on the influencing factors [46–49] of smart logistics, the opportunities and challenges faced by it [50,51], and the role of smart logistics in

enterprises, industries, countries [52,53], etc. From this point of view, the practice of smart logistics has made some achievements but still faces difficulties and challenges. Smart logistics is different from the traditional logistics industry. It applies smart technologies such as the Internet of Things, big data, cloud computing, and artificial intelligence to logistics. Therefore, it needs compound technical talents with both basic knowledge of logistics and smart logistics technology. Creating a good innovation environment is necessary to promote the high-quality development of smart logistics. In this section, we take China's smart logistics as an example and employ the above framework to study the mobility of China's smart logistics talents.

### 3.1. Data Acquisition

Invention patent data are conducive to understanding the research directions and involved fields of technical talents and possible contributions of these patents. Such information includes applicants' names, applicants' companies, the time of applications, the title of applications, etc., which can provide a foundation for tracking talents. Therefore, by referring to the latest literature, this paper determines a set of keywords [49,54–56] that integrate the essence of the supply chain and the characteristics of the digital economy. Smart logistics keywords include robot, cloud computing, Internet of Things, big data, RFID, radio frequency identification, GPS, positioning, navigation, obstacle avoidance, infrared remote sensing, blockchain, 5G, telecommunication, VR, virtual reality, simulation, simulation, sensor, scanner, AS/RS, stacking, access, Miniload, SCS, SSI SCHAEFER Rotary, smart tech, automation, AGV, unmanned technology, XML, database, integration, distribution, monitoring, intelligence, information, sorting, AR, stowage, packaging, pickup, Milkrun, POS, EDI, electronic data interchange, and GIS.

Based on the above keywords, this paper downloads 133,164 smart logistics invention patents from 2010 to 2021 through the Incopat patent platform. Before studying the mobility of China's smart logistics talents, this paper verifies the availability of the Incopat patent platform with recall ratio and precision ratio. We randomly select a smart logistics enterprise, such as SF Express. SF Express began to file patent applications in 2003, but until 2010, the number of its patent applications was small and was ignored here. According to the annual report of SF Express, 3112 items had been obtained and declared by the end of 2020, while 2701 patents were retrieved on the Incopat patent platform, with a recall ratio of about 86.8%. In addition, using "warehousing" as a keyword to search on the forum, 18 of the first 20 patent data points meet the purpose of retrieval, with a precision ratio of 90%. This shows that the Incopat patent platform has high reliability, which can be served as the data basis for discriminating smart logistics talents and reflecting their mobility.

The annual number and geographical distribution of such invention patents are shown in Figure 3.

As seen from the above figure, the number of invention patents in the field of smart logistics first displayed an increasing trend, rising from 1503 in 2010 to 25,618 in 2020. However, in 2021, due to the impact of COVID-19, invention patents decreased significantly to 16,446. Regarding geographical distribution, these patents were mainly concentrated in economically developed provinces along the eastern coast. The provinces/cities that have accumulated more than 10,000 invention patents include Beijing (21,574), Guangdong Province (21,383), Jiangsu Province (18,606), and Zhejiang Province (11,941).

The relationship between the flow scale and the growth rate of China's smart logistics talents in 2010–2021 is shown in Figure 4.

Noticeably, since 2010, the flow scale had shown an upward trend but remained relatively small before 2015, when less than 3000 talents relocated. It did not reach a peak until 2020 and then declined significantly for the first time in 2021. By contrast, the growth rate of talent flow decreased year by year from 2010 to 2014, remained stable from 2015 to 2018, then continued to decline since 2019, and turned negative in 2021. Overall, the growth rate of the flow of scientific and technological (S&T) talents shows an irreversible downward trend.

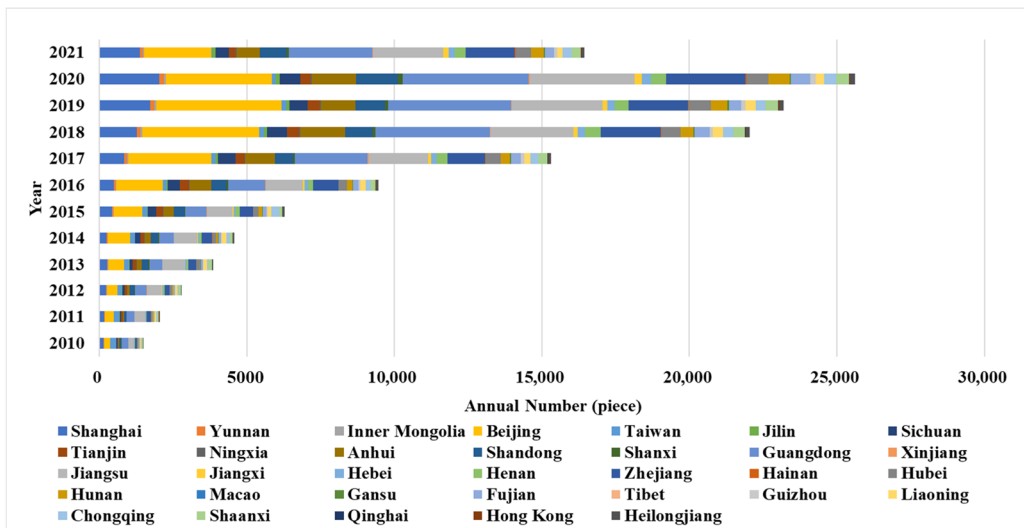

**Figure 3.** The number and geographical distribution of invention patent applications for smart logistics from 2010 to 2021.

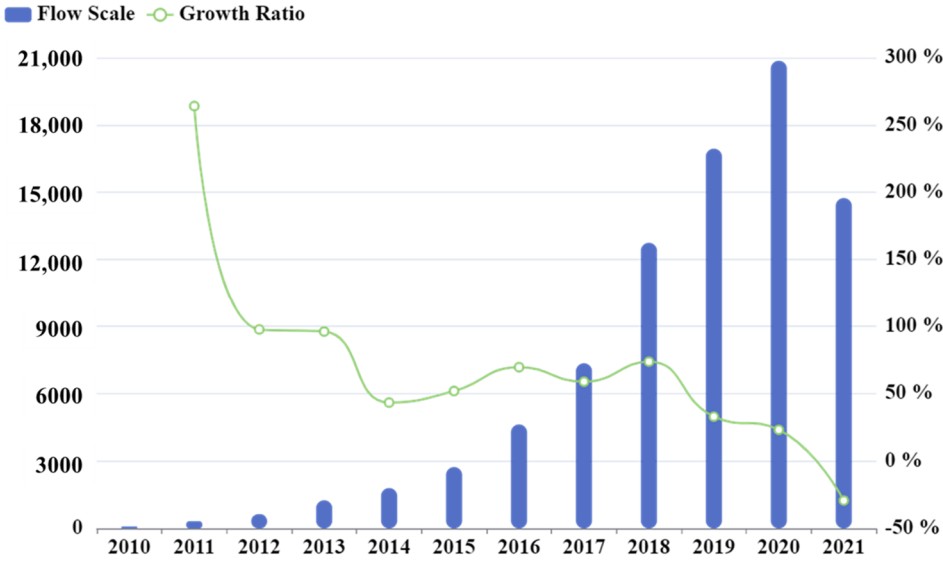

**Figure 4.** Combination chart of smart logistics talent flow scale.

### 3.2. Topology Diagram of the TTMN Model

This paper defines smart logistics talents as "technical personnel who participate in applying for smart logistics invention patents". Figure 5 shows the topology diagram of 12 TTMN models from 2010 to 2021.

In 2010, the TTMN model included 20 nodes, i.e., a total of 20 provinces involved in the inflow or outflow of smart logistics talents. In the following year, this number increased to 25 provinces, and then remained above 32 provinces since 2017, indicating that smart logistics talents have been migrating nationwide. Judging from the topological structure changes of the TTMN model, its network density shows an obvious rising trend year by year, indicating that the scope and frequency of flow of smart logistics talents are constantly increasing.

This paper introduces network density, global efficiency, average distance, average clustering coefficient, central potential and compatibility at the network level, and betweenness centrality at the node level and the edge level for a similar-weight network. We analyze the evolution variation trend of the topological structure of the TTMN model in

time series and study the intrinsic laws and evolution characteristics of talent flow in the field of smart logistics since the 18th National Congress of the Communist Party of China.

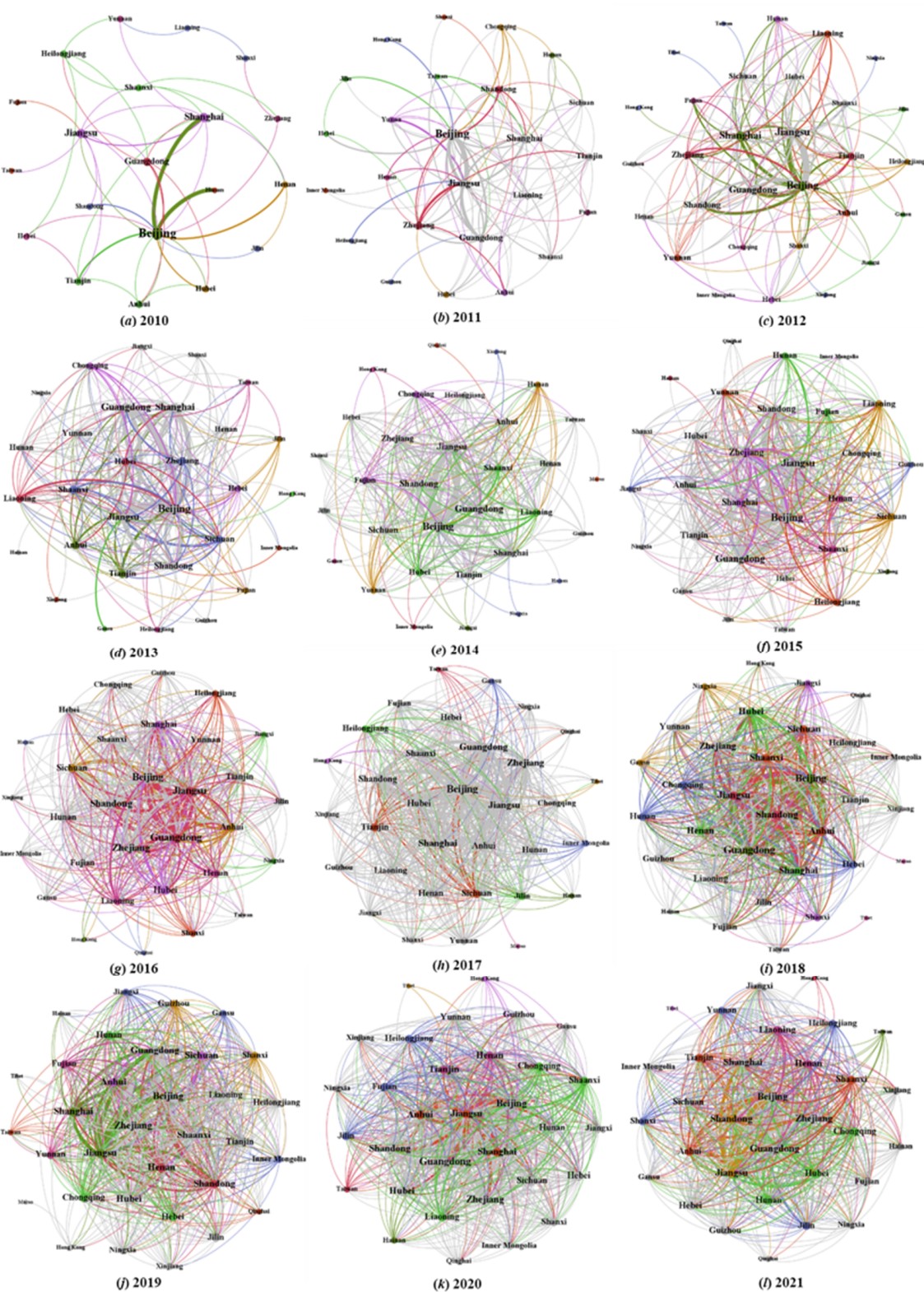

**Figure 5.** Topology diagram of the TTMN model. Note: The size of the node represents the degree, and the thickness of the edge represents the edge weight. The color of the edge is divided by the degree and is consistent with the source node.

### 3.3. Measurements of the TTMN Model

In order to show the change of the TTMN model in time series, the paper treats the TTMN model as an unweighted network and calculates the characteristic indexes at the network-level. The results are shown in Figure 6.

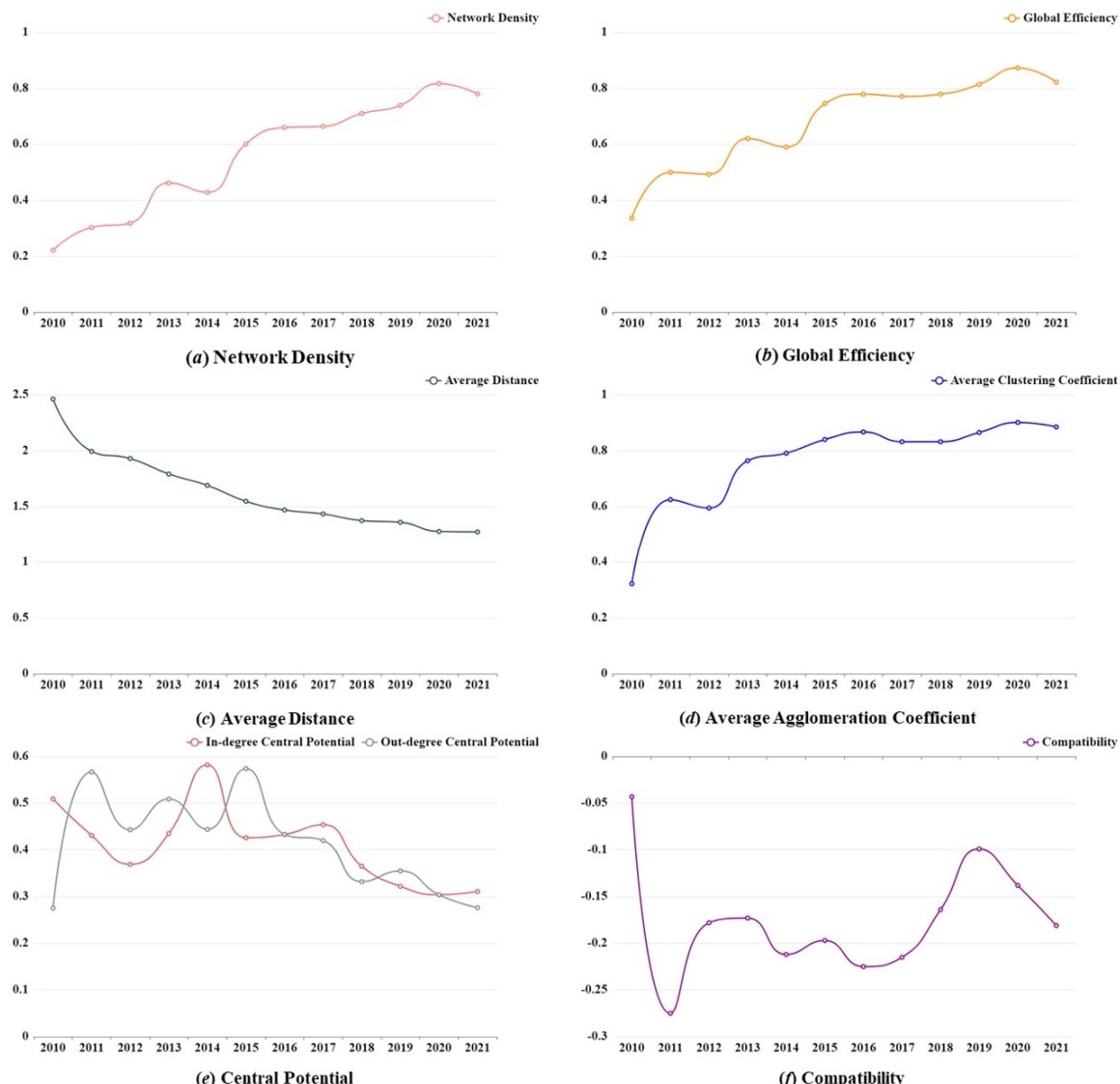

**Figure 6.** Network−level indicators of the TTMN model.

Through the statistical analysis of the above six network feature indicators, the following conclusions can be drawn.

First, network density showed an upward trend from 2010 to 2020 and a slight decline with negligible impact in 2021. This indicates that smart logistics talents are increasingly willing to relocate and have more and more diversified choices of workplaces, which has strengthened the ties between all provinces/cities and gradually gives rise to the sharing of technical knowledge.

Second, despite the fluctuation, global efficiency rose from 2010 to 2020. Economically developed provinces/cities have a siphon effect on talents, stimulating talents to flow; also, the flow of talents is also affected by national strategies and personal reality, showing the characteristics of regionally centralized flow. In short, the accelerated talent flow in the field of smart logistics between provinces and cities improves the wide attainment of knowledge and enhances the effect of technology diffusion.

Third, average distance steadily decreased from 2010 to 2021. For provinces and cities to obtain smart logistics technology directly or indirectly through talent flow, the difficulty has been lessened; the time has been shortened; the accuracy has been enhanced, which will reduce the loss of information in the process of technology diffusion and help to generate more innovations.

Fourth, the average agglomeration coefficient went up and down yet still had a great improvement in 2021 compared with 2010. This indicates an increased probability of smart logistics talents forming a closed-loop flow at the provincial and municipal level, closer cooperation between provinces and cities, and further industrial agglomeration. Combined with the change in average distance, the TTMN model presents an increasingly prominent small-world effect. Driven by industrial clusters, the nationwide technology diffusion efficiency in the field of smart logistics has been improved.

Fifth, in-degree central potential and out-degree central potential were not strongly correlated in the time frame, displaying radical changes. This demonstrates that the flow of smart logistics talents between provinces and cities needs to be balanced, which is closely related to the development level of the logistics industry in a specific province or city. In addition, it can be seen that in-degree central potential and out-degree central potential have shown a downward trend in recent years, indicating that the stability of the TTMN model has been improved, and the industrial layout has been made more reasonable.

Sixth, an all-time negative and oscillating trend could be noticed regarding compatibility from 2010 to 2021. This disassortative network, to a certain extent, indicates the decentralized characteristics of the TTMN model, which may be related to the different levels of appeal and prospects of the logistics industry in various provinces and cities. We believe that regions with relatively less advanced industrial development and insufficient talents reserve are more motivated to introduce stimulus policies and proactive talent attraction measures, which makes them the so-called "industrial depressions"—the popular destinations of talent flow in a specific period.

Then, we measure the TTMN model by using the node-level indicators. The node betweenness centrality of the TTMN model reflects the transfer hub function of a province/city on the transfer path of smart logistics talents. In other words, provinces/cities with higher $C_B^{RFWA}$ values have many input and output channels for smart logistics talents. According to the structural hole theory, such provinces and cities have two advantages. The first is the information advantage; that is, they have edges in technologies in this field through a large inflow of smart logistics talents. The second is the control advantage; that is, they can retain outstanding talents and eliminate those outside the process of industrial technology iteration. These two advantages complement each other so that these provinces and cities can achieve the Matthew effect of talent development. Table 2 lists the top five provinces and cities in terms of the number of node centrality from 2010 to 2021.

**Table 2.** Node betweenness centrality of the TTMN model.

| Year | No. 1 | | No. 2 | | No. 3 | | No. 4 | | No. 5 | |
|------|----------|---------------|----------|---------------|----------|---------------|----------|---------------|----------|---------------|
| | **Province** | $C_B^{RFWA}$ | **Province** | $C_B^{RFWA}$ | **Province** | $C_B^{RFWA}$ | **Province** | $C_B^{RFWA}$ | **Province** | $C_B^{RFWA}$ |
| 2010 | Beijing | 200 | Shanghai | 181 | Jiangsu | 123 | Guangdong | 71 | Shaanxi | 60 |
| 2011 | Beijing | 298 | Jiangsu | 243 | Zhejiang | 135 | Guangdong | 120 | Shanghai | 112 |
| 2012 | Beijing | 470 | Jiangsu | 302 | Shanghai | 280 | Guangdong | 105 | Tianjin | 82 |
| 2013 | Beijing | 711 | Guangdong | 175 | Jiangsu | 170 | Shaanxi | 142 | Shanghai | 123 |
| 2014 | Beijing | 741 | Guangdong | 294 | Jiangsu | 246 | Shanghai | 90 | Taiwan | 90 |
| 2015 | Beijing | 796 | Jiangsu | 202 | Anhui | 119 | Guangdong | 117 | Shanghai | 116 |
| 2016 | Beijing | 850 | Jiangsu | 157 | Guangdong | 153 | Zhejiang | 151 | Shandong | 93 |
| 2017 | Beijing | 920 | Guangdong | 157 | Jiangsu | 144 | Shanghai | 97 | Zhejiang | 96 |
| 2018 | Beijing | 879 | Guangdong | 269 | Shaanxi | 155 | Shandong | 96 | Sichuan | 94 |
| 2019 | Beijing | 815 | Guangdong | 270 | Jiangsu | 248 | Shanghai | 156 | Shandong | 127 |
| 2020 | Beijing | 784 | Jiangsu | 242 | Guangdong | 235 | Zhejiang | 95 | Shanghai | 64 |
| 2021 | Beijing | 709 | Guangdong | 274 | Jiangsu | 246 | Shandong | 91 | Shanghai | 62 |

Through the statistical analysis of node betweenness centrality, this paper draws the following conclusions.

First, on the whole, even among the top five provinces and cities, there are apparent differences in the node betweenness centrality. Being ahead of other provinces/cities every year, Beijing is located at the core of the TTMN model as a smart logistics talents transfer hub with absolute talent competitive advantages and the most robust control over other provinces/cities, presenting the dynamic of "one superpower and multi-great powers". With its obvious geographical advantages, Beijing boasts leading S&T resources and a sound environment for developing the logistics industry. Beijing is not only the center for four significant logistics and distribution methods but also is the base of many leading logistics companies, which together bring the advantages of the cluster effect to the development of Beijing's logistics industry, making Beijing an important talent pathway.

Second, Guangdong Province and Jiangsu Province have relatively high node betweenness centrality, taking the second and third positions in turns. Although they are less competitive than Beijing in smart logistics talent transfer ability, these two provinces are in a relatively core and pivotal position in the TTMN model. Guangdong Province serves as a transportation hub in the Pearl River Delta region, with a perfect network, many high-quality ports with considerable throughput, and an advanced economic level, promoting the logistics industry's rapid development. In the Yangtze River Delta region, Jiangsu Province is at the intersection of the Belt and Road. With frequent inbound and outbound trade, it attracts investment from domestic and foreign enterprises [57], which expedites the expansion of its logistics industry. Therefore, the two provinces' logistics industry has relatively strong competitiveness, making them a critical "transfer station" for logistics talents.

Third, Shanghai ranked around fourth or fifth in most years except for 2010 and 2012 and dropped out of the top five several times. From the view of the numerical value of node betweenness centrality, Shanghai has been lagging behind Beijing, Guangdong Province, and Jiangsu Province. The difference between Shanghai and Beijing has been more than ten times in recent years. To some extent, this shows that the core position of Shanghai in the TTMN model has been challenged. In recent years, limited by the scale effect, Shanghai's economic and information advantages have yet to be fully reflected. In addition, the increase in logistics operating costs for labor and land has weakened the logistics industry's comprehensive competitiveness, the declining status as a transit hub for smart logistics talents, and the gradual loss of its competitive advantages compared with Beijing, Guangdong Province, and Jiangsu Province.

Fourth, the node betweenness centrality of other provinces/cities, except for those mentioned in Table 2, is relatively small. These provinces/cities are located at the edge of the TTMN model and are less affected by Beijing and other core provinces/cities. They need more muscular control over other provinces/cities and insufficient talent exchanges with each other and are thus not capable of being talent transfer hubs.

Finally, we measure the TTMN model by using the edge-level index. Unlike the node betweenness centrality, the edge betweenness centrality in the TTMN model measures the transfer function of the smart logistics talent exchanges between two provinces/cities for the overall connectivity of the network. It bears different meanings from the frequency of smart logistics talent exchanges. It integrates the indirect impact on the overall network, thereby better reflecting the importance of the specific flow of smart logistics talents. Figure 7 is a heatmap based on $C_E^{RFWA}$ values, and its value is calculated by the cumulative scale of China's smart logistics talents from 2010 to 2021.

As can be seen from Figure 7, both inward and outward flows of smart logistics talents between Beijing and other provinces play a pivotal role. In addition, some inter-provincial flows are also essential, including Jiangsu Province → Taiwan ($C_E^{RFWA} = 65$), Taiwan → Macao ($C_E^{RFWA} = 33$), Shanghai → Hong Kong ($C_E^{RFWA} = 32$), Taiwan → Guangdong Province ($C_E^{RFWA} = 32$), and Macao → Zhejiang Province ($C_E^{RFWA} = 31$).

To display the important flows of smart logistics talents in China, we extract the edges whose $C_E^{RFWA} > 27$ from the TTMN model to form a new network, as shown in Figure 8.

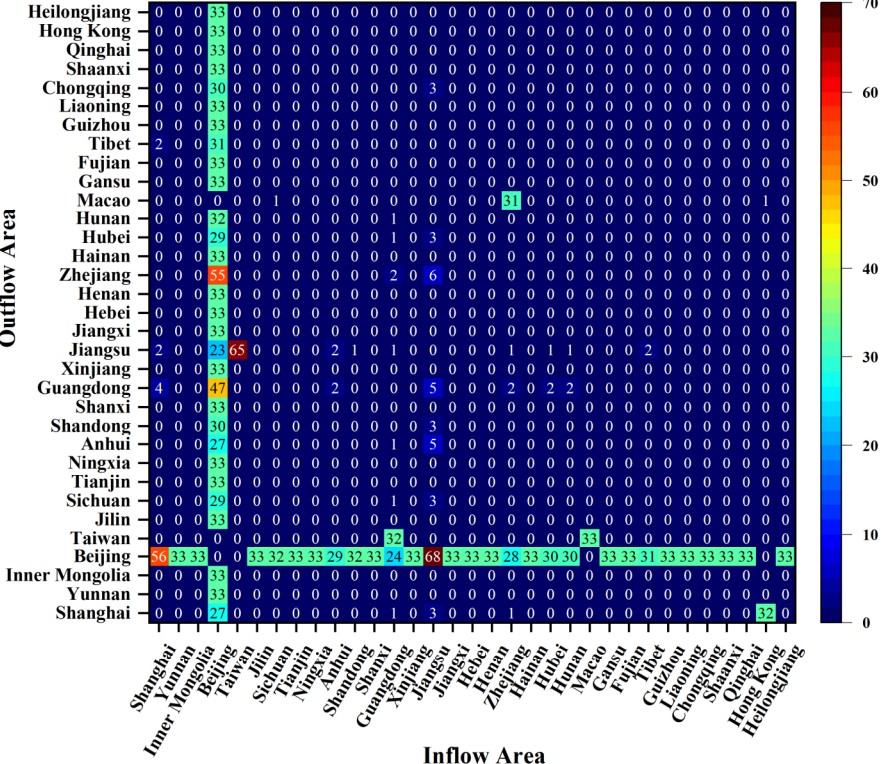

**Figure 7.** Edge betweenness centrality of the TTMN model.

Through the statistical analysis of the betweenness centrality, the following conclusions can be drawn.

First, this network resembles a star network. Beijing is positioned in the center of the network, in and out of which there are edges connecting other provinces/cities, indicating that the smart logistics talent flow between Beijing and other provinces/cities is overall mutual, which further confirms that Beijing acts as a vital hub for the transfer of smart logistics talents. Beijing enjoys a well-developed logistics industry with cutting-edge logistics infrastructure, large-scale and highly specialized logistics enterprises, and strong talent attraction, which has driven technological innovation and industry progress. In addition, the two-way promotion of talents and technology forms a virtuous circle, and the technology diffusion and spillover effects are significant. Through the bridge of Beijing, smart logistics talents can flow nationwide.

Second, regarding the level of interaction, Beijing has the closest ties with Shanghai, Guangdong Province, Jiangsu Province, and Zhejiang Province. Among them, Shanghai and Jiangsu Province are essential recipients of smart logistics talent from Beijing, and Guangdong Province and Zhejiang Province are important senders of smart logistics talent to Beijing. These five provinces/cities are located in the Beijing–Tianjin–Hebei urban agglomeration, the Yangtze River Delta urban agglomeration, and the Pearl River Delta urban agglomeration, which have unique geographical advantages and a full-fledged logistics industry. At the same time, there is a general spillover effect regionally, and the talent interaction between them is relatively more frequent.

Third, other key pathways include Jiangsu Province→ Taiwan ($C_E^{RFWA}$ = 65), Taiwan → Macao ($C_E^{RFWA}$ = 33), Shanghai → Hong Kong ($C_E^{RFWA}$ = 32), Taiwan → Guangdong Province ($C_E^{RFWA}$ = 32), and Macao → Zhejiang Province ($C_E^{RFWA}$ = 31). These provinces/cities are all situated in the eastern part of China with geographical proximity, which is conducive to and convenient for talent exchange. Moreover, all these

provinces/cities have a relatively high level of economic development and enjoy sound economic and trade ties with each other, further promoting the flow of talent.

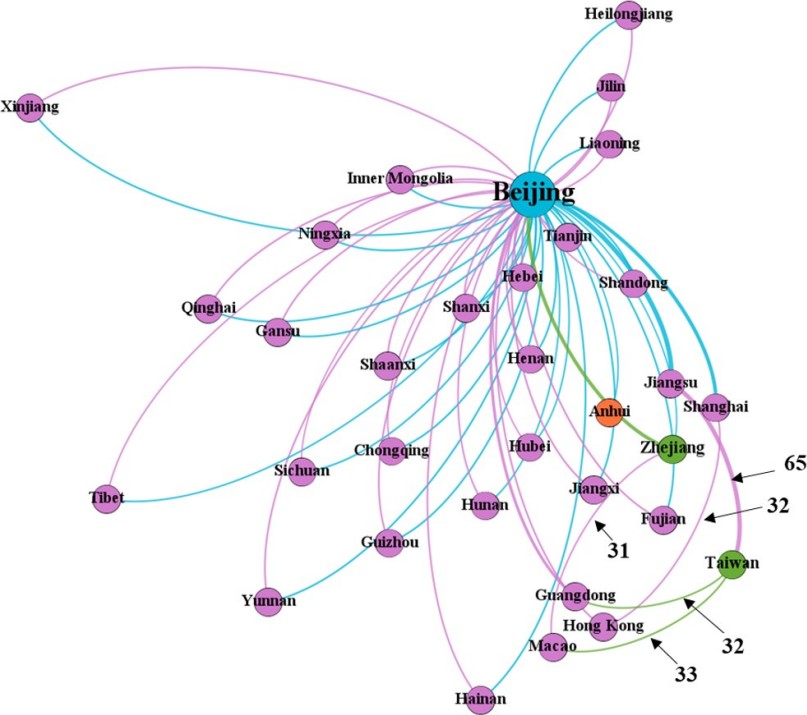

**Figure 8.** Important channels of smart logistics talent transfer in China. Note: The size of the node represents the degree, and the thickness of the edge represents the edge weight. The color of the edge is divided by the degree and is consistent with the source node.

## 4. Discussion and Conclusions

Generally, when studying the mobility of scientific talents, their mobility depends on the changes in their affiliated institutions in the data source. However, this method only applies to technical talents and has a specific time lag and cumbersome processing. Considering that technical talents are practical personnel, this paper focuses on their invention patent data. It determines their liquidity with the change of applicants, which can reduce the difficulty of data acquisition and improve data integrity to a certain extent. In addition, in this paper, we establish a comprehensive network framework to describe the mobility of technical talents. The framework used in this paper can not only depict a more complete picture of China's smart logistics talent mobility but also provide a valuable reference for studying talent mobility in many other technical fields.

It is found that under the framework proposed in this paper, the current situation of talent mobility in the field of China's smart logistics has been better described. For example, the network density, global efficiency, and average agglomeration coefficient of the TTMN model show an overall upward trend, while the average distance shows a downward trend. The in-degree central potential and out-degree central potential fluctuate, with the compatibility always being negative. It reveals that talent mobility brings about an increasingly close relationship between provinces/cities, a more prominent technology diffusion effect, and a more balanced resource distribution. More obviously, Beijing is at the core of the TTMN model, which has the most talent exchange channels, represented by those Shanghai, Guangdong Province, Jiangsu Province, and Zhejiang Province. In the context of the new scientific and technological revolution, as a cross-border integration industry, smart logistics has become a new trend in the development of the logistics industry, which has been highly valued by the state and mentioned many times in policy documents such as the "Made in China 2025". The conclusions are consistent with the actual situation [58–60]. Nevertheless, previous research tends to have a statistical or qualitative

description, lacks a clear measurement framework, and has little focus on the network characteristics. The framework proposed in this paper makes up for these imperfections, which can comprehensively depict the mobility of China's smart logistics talents.

This study is a preliminary exploration of the mobility of technical talents from the perspective of the complex network. It provides a reference for encouraging the rational flow of talents and taking full advantage of human resources, but there are also some limitations. First of all, the data source is the Incopat patent platform, which does not rule out data omissions, such as some patents that have yet to be retrieved. Future research can combine other patent search platforms to acquire richer data. Secondly, this study only analyzes China's smart logistics field empirically. To make the research framework more universal, the combination of quantitative methods and qualitative methods can be considered for comparative analysis in the future.

**Author Contributions:** Conceptualization, J.G., C.L., G.L. and L.X.; methodology, J.G., C.L. and L.X.; software, C.L. and L.X.; formal analysis, C.L., G.L. and L.X.; investigation, J.G. and L.X.; writing—original draft preparation, C.L. and L.X.; writing—review and editing, J.G., C.L., G.L. and L.X.; visualization, C.L.; supervision, J.G., G.L. and L.X.; funding acquisition, G.L. All authors have read and agreed to the published version of the manuscript.

**Funding:** This research was funded by the National Natural Science Foundation of China (Grant No. 72204014) and the Beijing Social Science Fund Project (Grant No. 21JCC051).

**Institutional Review Board Statement:** Not applicable.

**Informed Consent Statement:** Not applicable.

**Data Availability Statement:** The raw datasets used in this study can be obtained by contacting the corresponding author.

**Conflicts of Interest:** The authors declare no conflict of interest.

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
