# Peer review of "Framework to Measure the Mobility of Technical Talents: Evidence from China’s Smart Logistics"

_sustainability, doi:10.3390/su15032481_

Round 1

Reviewer 1 Report

The proposal about “Framework to Measure the Mobility of Technical Talents: evidence from China’s Smart Logistics” sounds well. I suggest some questions to add, optionally. You need the implement the next questions:

-Title. It is right.

-Abstract. It is right.

-Theoretical framework. Check if some references can be updated. To be prudent, try to update some new references if you find.

-Methods. The used method is right. Optionally, you can add a quantitative tool. The paper is positive like now, but I suggest to complete the paper with other qualitative sources, for example in-deep interviews or a Delphi. It means it is positive to do a triangulation to support the obtained data. Perhaps in-deep interviews are not possible in this moment, but expert brief interviews (a Delphi) can allow an added value. It is a recommendation.

-Results. Results are right. If you decide to develop the qualitative added method, you can adapt it.

-Conclusion and discussion. Try to enlarge it if you include another method. If not, you can maintain it as now.

Reviewer 2 Report

The work is very current and interesting.

The structure of the work is well developed and a clear research methodology is presented.

I support the authors in researching this topic.

Reviewer 3 Report

Dear Sir,

After review of the paper titled "Framework to measure the mobility of technical talents: Evidence from China’s smart logistics" which submitted to the sustainability, I think that this paper is very good. This paper contains many important and significant empirical contributions which can be used by academics, researchers and policy makers.

All the best
